# The Toxicity and Polymorphism of β-Amyloid Oligomers

**DOI:** 10.3390/ijms21124477

**Published:** 2020-06-24

**Authors:** Ya-ru Huang, Rui-tian Liu

**Affiliations:** 1State Key Laboratory of Biochemical Engineering, Institute of Process Engineering, Chinese Academy of Sciences, Beijing 100190, China; yrhuang@ipe.ac.cn; 2School of Chemical Engineering, University of Chinese Academy of Sciences, Beijing 100049, China

**Keywords:** β-amyloid oligomers, Alzheimer’s disease, polymorphism, toxicity, aggregation

## Abstract

It is widely accepted that β-amyloid oligomers (Aβos) play a key role in the progression of Alzheimer’s disease (AD) by inducing neuron damage and cognitive impairment, but Aβos are highly heterogeneous in their size, structure and cytotoxicity, making the corresponding studies tough to carry out. Nevertheless, a number of studies have recently made remarkable progress in the describing the characteristics and pathogenicity of Aβos. We here review the mechanisms by which Aβos exert their neuropathogenesis for AD progression, including receptor binding, cell membrane destruction, mitochondrial damage, Ca^2+^ homeostasis dysregulation and tau pathological induction. We also summarize the characteristics and pathogenicity such as the size, morphology and cytotoxicity of dimers, trimers, Aβ*56 and spherical oligomers, and suggest that Aβos may play a different role at different phases of AD pathogenesis, resulting in differential consequences on neuronal synaptotoxicity and survival. It is warranted to investigate the temporal sequence of Aβos in AD human brain and examine the relationship between different Aβos and cognitive impairment.

## 1. Introduction

Alzheimer disease (AD) is a chronic neurodegenerative disease which imposes a heavy burden on families and society [1]. The neuropathological profiles of AD are mainly manifested as the extracellular senile plaques aggregated by β-amyloid (Aβ), and the formation of intracellular neurofibrillary tangles aggregated by hyperphosphorylated tau protein [2]. Aβ has normal physiological function and maintains a low concentration in vivo [3], however, factors such as aging, oxidative stress and gene mutation cause the disruption of Aβ homeostasis, resulting in Aβ accumulation and aggregation, formation of oligomers and fibers, and plaque deposits in brains [4]. Heavy evidence has confirmed that Aβ oligomers (Aβos) are the most neurotoxic aggregates and play a critical role in the occurrence and development of AD by causing functional neuron death, cognitive damage, and dementia. In recent years, Aβos have been extracted and characterized by various methods, and remarkable progress has been made in the mechanism of pathogenesis. In this review, we introduce the neurotoxic mechanisms of Aβos, and review the properties of Aβos with different size, conformation and neurotoxicity.

## 2. The Formation of Aβos

Aβ is produced principally in neuronal endosomes by hydrolysis of amyloid precursor protein (APP) with β-secretase and γ-secretase [5,6], and its release is modulated by synaptic activity [7]. Studies identified that under normal physiological conditions, Aβ can inhibit the excessive activation of synapses and reduce the production of synaptic excitotoxicity [8,9,10]. Moreover, Aβ may be an antibacterial peptide in CNS innate immunity, which protects the body from infections, and Aβ aggregation may be a protective response against infection [11,12]. But under the pathological condition, the production of Aβ increases or the clearance is inhibited, leading to Aβ abnormal accumulation and aggregation to form β-sheet-rich conformations [13,14].

Oligomers are soluble polymers assembled by monomers, some of which are intermediates in fibril generation. Oligomers have different molecular weights and conformations, which results in various oligomeric properties. There is plenty of evidence that Aβos contribute to the development of AD, therefore, clarifying the molecular events related to the aggregation of Aβ monomers into toxic Aβos is the key to understand the pathological mechanism of AD. Aβ aggregates can formed by the assembly of two or more monomers through fibril-independent pathway (primary nucleation) or fibril-dependent mechanism (secondary nucleation), in which Aβ monomers aggregate on the surface of fibrils in a prion-like manner [15,16]. Monitoring the kinetics of Aβ aggregation in vitro with synthetic Aβ, the process is typically divided into three phases: an initial lag phase, a subsequent growth phase and a final stationary phase [17] (Figure 1). 

Aβos are intermediates of Aβ aggregation, which are formed in lag phase and growth phase and reveal a range of intermediates with different size, morphology and conformation [18,19]. Some oligomeric Aβ intermediates such as Aβ dimers, denoted fibrillar oligomers, enter the fibril-forming pathway to form fibrils, whereas others such as Aβ trimers or globulomers pool as stable non-fibrillar oligomers [20,21]. Latest study showed that non-fibrillar oligomers could convert into fibrillar oligomers but the rate of oligomer conversion was very slow [16]. Aβos were first detected in 1994. Researchers extracted the soluble Aβ component in the brain by a non-denaturing method, and found that Aβ level in the soluble component was related to the severity of AD [22,23]. The discovery of Aβos and the inconsistency between cognition and brain plaques shifted the focus of Aβ research from fibers to oligomers.

## 3. The Neurotoxicological Mechanisms of Aβos

The “amyloid cascade hypothesis”, proposed in 1992, suggests that Aβ deposition is the first step in the pathological development of AD, which then leads to tau pathologies, synaptic dysfunction, neuron loss and dementia [24]. Subsequently, due to the significant toxicity of Aβos, and the close relationship between oligomers and cognition and memory, the hypothesis was revised to “Aβos cascade hypothesis” [25], which suggests that soluble Aβos directly cause neural signal dysfunction, induce neuronal apoptosis, and inhibit synaptic long-term potentiation (LTP). Multiple studies have shown that Aβos exert neurotoxic effects through multiple mechanisms, such as receptor binding, mitochondrial dysfunction, Ca^2+^ homeostasis dysregulation and tau pathologies (Figure 2).

### 3.1. Receptor-Mediated Neurotoxicity of Aβos

Aβos can bind to the receptors on the surface of neurons, alter downstream signaling pathways, and cause cell death ultimately. There are more than 20 types of receptors recognized by Aβos such as cellular prion protein (PrP^c^), glutamate receptors, β_2_-adrenergic receptor (β_2_-AR), p75 neurotrophin receptor (p75^NTR^) and α7-nicotinic acetylcholine receptor (α7nAChR) [26]. Under normal physiological conditions, PrP^c^ is an important protein regulating Aβ metabolism in vivo by inhibiting β-secretase activity and decreasing the production of APP intracellular domain (AICD) [27]. However, Aβos bind to PrP^c^ and affect PrP^c^ normal physiological functions in AD brains [28,29]. Aβos can directly interact with PrP^c^, impair LTP and cause neuritic dystrophy by activating Fyn tyrosine kinase and causing phosphorylation of *N*-methyl-d-aspartate receptor (NMDAR), but when the expression of PrP^c^ is removed, these adverse effects are deprived [30,31,32].

Aβos can impair the activity of glutamate receptors to interfere with synaptic plasticity [33]. NMDAR is a glutamate receptor, which has ion channel activity, and plays a critical role in regulating synapse formation and synaptic plasticity [34]. Some studies have indicated that Aβos disturbed NMDAR activity and impaired NMDAR-dependent signaling pathway, leading to synaptic loss and the decrease of spinal density [7,35]. In addition, Aβos disturb the NMDAR-mediated postsynaptic Ca^2+^ signaling by affecting the function of NMDAR and altering the availability of extracellular glutamate, which causes LTP impairment [36]. α-Amino-3-hydroxy-5-methyl-4-isoxazolepropionic acid receptor (AMPAR) is another glutamate receptor containing four subunits GluA1-A4, which participates in modulation of synaptic plasticity and homeostasis [37]. Aβos induce the reduction of AMPAR level by accelerating the degradation, and increase AMPAR ubiquitination, resulting in the loss of AMPAR activity and suppression of synaptic transmission [38]. Moreover, Aβos cause tau hyperphosphorylation and abnormal accumulation in dendritic spines, which in turn deficits the Aβos-mediated AMPAR signaling pathway, forming a vicious cycle [39].

β_2_-AR, expressed in hippocampus and cortex, is required for learning and memory. Soluble Aβ42 can bind to the extracellular N terminus of β_2_-AR and then interfere with AMPAR activity by inducing G-protein/cAMP/protein kinase A (PKA) signaling, which may cause excessive Ca^2+^ to flow into neurons and induce excitatory neurotoxicity, leading to overactivity of neurons and loss of dendritic spines ultimately [40,41]. Research has found that low molecular weight (MW) Aβos led to the decrease of β_2_-AR level and impairment of hippocampal LTP [42]. In addition, β_2_-AR can promote tau protein phosphorylation through the β_2_-AR-PKA-JNK signaling pathway [40]. These results demonstrate that β_2_-AR may have an important effect on AD pathogenesis.

p75^NTR^ is a transmembrane protein, which participates in Aβos-mediated neurodegenerative signals. In cholinergic basal forebrain (cBF) neurons, Aβos bind to cysteine-rich repeat domains 2 and 4, the extracellular domain of p75^NTR^ [43], to mediate tau pathology, synaptic disorder and promote the degradation of neurons [44]. Removal of p75^NTR^ expression from cBF neurons causes the improvement of Aβ plaque deposition and cognitive impairment and attenuates the tau pathologies in AD transgenic mice [45,46]. Moreover, after binding to Aβ, p75^NTR^ can regulate Aβ production by interacting with beta-site amyloid precursor protein cleaving enzyme-1 (BACE1) and enhancing the colocalization and internalization of BACE1 and APP [47], besides, it also has an impact on the clearance and deposition of Aβ [48]. The latest study identified that p75^NTR^ influenced the effect of Aβ42 on the L-type Ca^2+^ channel current by disturbing the function of voltage-dependent Ca^2+^ channels, which can be eliminated after decreasing the expression of p75^NTR^ [49].

Aβos have high affinity with α7nAChR [50], and lead to the internalization of the Aβ–α7nAChR complex, which induces plaque formation and promotes neurotoxicity [51,52]. However, some studies have found that α7nAChR had a neuroprotective effect. Activation of α7nAChR inhibited neuronal γ-secretase activity, and p38 or JNK activity, both of which contribute to the alleviation of Aβ burden and Aβ-induced memory deficits [53,54,55].

In addition to the above receptors, Aβos can also bind to paired immunoglobulin-like receptor B (PirB) [56,57], EphB2 receptor [58,59], metabotropic glutamate receptor 5 (mGluR5) [60,61] and other receptors to exert neurotoxicity. However, it is not explicit that which receptor has the highest affinity to Aβos in brains or induces deadly damage to neurons. Aβos are highly heterogeneous with multiple sizes and conformations, and different oligomers may bind to different receptors. The studies to determine the binding of certain Aβ oligomer to its receptors to exert specific toxicity are warranted.

### 3.2. Cell Membrane Destruction

Aβos rather than monomers and fibers can directly act on the cell membrane to form channels, and destroy membrane integrity and permeability [62,63,64]. High-MW aggregates elevate the intracellular Ca^2+^ level by increasing the membrane lipid peroxidation reaction and changing the membrane structure and function, resulting in the suppression of LTP and neuronal death [65]. On the other hand, after binding to the β-sheets of the N-terminal residues of Aβos, lipid bilayers stabilize the structure of Aβos and accelerate fiber formation [66]. The molecular docking results showed that the U-shaped oligomers inserted into the membrane and formed ion-permeable pores [67], which is consistent with the report that Aβ can insert into lipid bilayers and assemble into a β-barrel pore under optimized detergent micelle conditions [68]. Aβos can also interact with lipids and cholesterol in the membrane to exert their toxicity [69,70,71,72,73]. For example, Aβos strongly bind to GM1 ganglioside located at the external face of the cell membrane to promote LTP impairment, which requires the existence of sialic acid residues of GM1 [71]. Moreover, GM1 not only affects Aβos-mediated perforation of cell membrane [73], but also influences the aggregation and the secondary structure of Aβos [70,74]. The effect of cholesterol on the toxicity of Aβos to membrane is controversial. Cholesterol can change the structures of Aβos and facilitate the interaction between Aβ and the membrane, on the other hand, cholesterol increases the membrane rigidity and has a protective effect against Aβos-mediated damage [69,72].

### 3.3. Mitochondrial Damage and Ca^2+^ Homeostasis Dysregulation

Aβos increase Ca^2+^ influx and cause intracellular Ca^2+^ homeostasis dysregulation by binding to receptors on neuronal membranes and disrupting the structure and permeability of cell membranes [75,76]. Increased intracellular Ca^2+^ induces the activization of the Ca^2+^/calmodulin-dependent phosphatase calcineurin (CaN) and glycogen synthase kinase 3β (GSK3β), which can induce hyperphosphorylation of tau, impair the transport of brain-derived neurotrophic factor (BDNF) and cause transport dysregulation in both axons and dendrites [77]. Moreover, the destruction of intracellular Ca^2+^ homeostasis causes mitochondrial Ca^2+^ overload and mitochondrial dysfunction [78,79]. The mitochondrial Ca^2+^ dysregulation then directly induces superoxide generation, ATP utilization and neuron death by increasing the opening of mitochondrial permeability transition pores (MPTP) and activization of Ca^2+^-dependent proteases (calpains) [80,81]. In addition, Aβos elevate parkin level in mitochondria and induce mitochondrial dysfunction, but these adverse effects are eliminated when parkin is overexpressed [82]. Mitophagy is an essential cell activity to degrade and remove damaged mitochondria or excess mitochondria, which participates in the adjustment of the functional integrity of the mitochondrial network and cell survival [83]. Mitochondrial Ca^2+^ dysregulation also interferes with mitophagy [84], which further aggravates the generation of Aβ aggregates and AD pathological development, forming a vicious cycle between Aβos and mitochondrial disability [85,86].

### 3.4. Tau Pathologies

Aβ and tau are two main pathogenic factors in the pathogenesis of AD. Although there is no clear answer which is the key factor responsible for the onset of AD, some progress in our understanding of the relationship between Aβ and tau has been achieved in recent years. Longitudinal evaluations of Aβ and tau PET imaging have indicated that Aβ deposition first appears in the cortex, at least 10–20 years earlier than clinical symptoms, which precedes the development of tau lesions, neuron death and hippocampal volume loss [87,88,89]. The rate of Aβ accumulation can predict the occurrence of tau aggregation, and the rate of tau accumulation is associated with decreased global cognition, which can predict the severity of cognitive impairment [90]. Examining the interaction between Aβ and tau indicated that Aβ deposition may be used as seeds to facilitate the pathological development of tau by creating a unique environment [91,92,93], which is coincident with the research that tau aggregates could not widely spread in the absence of Aβ deposition [94]. In addition, Aβos bind to α2AAR receptors on the surface of neurons, thereby activate GSK3β and promote tau hyperphosphorylation. The required concentration of Aβos to promote tau hyperphosphorylation through this pathway is much lower than that to play direct effects [95]. However, latest study showed that tau burden first emerged in entorhinal cortex and hippocampus of AD patients, which preceded the global cortical Aβ by about 3 years [96].

## 4. The Polymorphism of Aβos

Aβos are highly heterogeneous and exhibit diversities in size, conformation, aggregation mode, toxicity and appearance time in the brain. In order to accurately explore the characteristics of different Aβos, the first step is to separate oligomers with different sizes or conformations. Currently, some approaches such as centrifugation and size exclusion chromatography (SEC) can separate oligomers according to molecular size but cannot distinguish conformations. Multiple Aβos-specific antibodies such as A11, OC, as well as W20, NAbs-Aβo, prepared in our laboratory, have been confirmed to recognize the advanced conformation of Aβos [97,98,99,100], which can be used to extract Aβos with different conformations. Separated by these strategies, the properties of different Aβos can be studied detailly. Next, we review the characteristics and toxicity of several main types of oligomers, including dimers, trimers, Aβ*56, and spherical oligomers (Table 1).

### 4.1. Aβ Dimers

Aβ dimers are the smallest Aβ aggregates, and they have three characteristics: high levels in the brains of AD patients and transgenic mice [121,122,123,124], stable in SDS and strong denaturants [101,104,122,125,126], and significant neurotoxicity [101,102,103,104]. When treated with 6.8M guanidine thiocyanate, the soluble Aβ aggregates and insoluble Aβ plaques in AD brains are dispersed into two bands in SDS gel electrophoresis (SDS-PAGE), such as Aβ monomers and dimers [104,124,125]. It is proposed that Aβ dimers are formed by cross-link via tyrosine acid residue phenol coupling (DiY) and enzymatic coupling (QK) between glutamine and tyrosine residues, resulting in the resistance of dimers to the strong denaturants [125,126]. The latest study showed that Aβ dimers derived from AD human brains were composed of multiple Aβ monomers with different lengths, and these different monomers were covalently coupled at the ASP1 and Glu22 sites to form stable dimers [101]. However, it is unclear whether the toxicities of the dimers formed through different cross-linking sites are different.

It is generally considered that Aβ dimers may be the basic constituent unit of fibers or oligomers. In Tg2576 and J20 model mice, Aβ dimers were detected after the age of 10 months, in which a large number of plaque deposits appeared in the cortex, indicating that there may be some connections between Aβ dimers and fibers [122,127]. Moreover, Aβ dimers are generally observed in AD patients over 60 years old. The level of Aβ dimers subsequently increases sharply, which shows the positive correlation between dimers and plaque in the brains [110]. In addition, soluble Aβos in the brain extracts of AD patients were broken down into two molecules of 6.1–7.9 kDa and 3.5–4.5 kDa, when treated with strong denaturants. It is thus speculated that Aβ dimers may also be the basic unit of Aβos [124].

Adding brain-derived Aβ dimers to primary neurons can reduce the length of neurons [101], trigger synaptic damage and neuron death, inhibit LTP response and induce tau hyperphosphorylation [102]. In tgDimer mouse, Aβ dimers contribute to the reduction of hippocampal acetylcholine and serotonin turnover rates, resulting in neurotransmitter dysfunction and behavioral impairments [103]. The latest study identified that brain-derived Aβ dimers caused neuronal overactivation by inhibiting the glutamate reuptake process. This process occurs earlier than clinical symptoms and plaque formation [104], indicating that Aβ dimers may contribute to the early development of AD.

### 4.2. Aβ Trimers

Although there is no obvious correlation between the trimers and plaque deposits, Aβ trimers are considered to be an aggregation unit of multiple Aβos such as hexamers and dodecamers [128]. Studies have shown that Aβ*56 is a type of Aβos assembled by trimers [109] and the first step of amylospheroids (ASPD) formation is to from trimers [20]. Moreover, Aβ trimers are the earliest Aβ aggregates in the brains of AD human and transgenic mice. In the primary cortical neurons of Tg2576 mice, monomers and trimers can be detected before dimers, indicating that trimers may be produced at the early stage. Further research suggested that Aβ trimers first appeared in the embryonic stage of Tg2576 mice, then increased steadily with aging, and existed throughout life [109]. Consistent with AD animal models, in human brain, the trimers exist since childhood, and their levels gradually increase with age, but there is no significant correlation between trimers and plaque deposits [110].

The toxicity of Aβ trimers is currently controversial. It was found that trimers and tetramers prepared by PICUP aggregation had prominent toxicity to neurons, but the toxicity of dimers was very weak [105]. Moreover, reducing the amount of Aβ trimers in the brain of APP/PS1 mice significantly alleviated the cognitive impairment [106]. When the cultured primary neurons were treated with Aβ trimers isolated from brain tissues of AD patients, trimers induced the change of tau conformation and caused the disruption of axonal transport, both of which were associated with reduction of the kinesin-1 light chain (KLC1) [107]. However, some studies argued that Aβ trimers acted only as a fundamental assembly unit for toxic assemblies rather than a toxic element. They found that the trimers isolated from 7PA2 cells or AD mouse brains significantly inhibited the LTP response, but the toxicity was smaller than homodimers and Aβ*56 [108,129]. In addition, there is no clear relationship between Aβ trimers and the cognitive ability of AD mice [109], and the connection between trimers and hyperphosphorylation of tau protein in the brain of AD patients is also ambiguous [110].

### 4.3. Aβ*56

Aβ*56, first detected in Tg2576 mice, is consistent with the occurrence and development trend of cognitive impairment in mice [109]. Some studies have also reported the appearance of Aβ*56 in other AD transgenic mouse models, such as J20 mice [127,130], Arc6/48 mice [130], and 3xTgAD mice [131]. In AD human brains, the level of Aβ*56 is negligible in children and adolescents, but the level rises above baseline and then steadily rises after the age of 40 [110]. Aβ*56 can be recognized by oligomer-specific antibody A11, but not by fiber-specific antibody OC, indicating that Aβ*56 is a non-fibrous Aβ aggregate. Moreover, in brains of AD transgenic mice, Aβ*56 mainly locates on the plaque free tissue and halo, but not plaque core, suggesting that it may have a greater effect on cognitive impairment than other Aβos wrapped in the plaque core [97].

It is generally considered that Aβ*56 has conspicuous toxicity to neurons, and is related to cognitive impairment in a variety of AD transgenic mice [127,130,131]. The expression of Aβ*56 in olfactory epithelium significantly increases in the early onset of Tg2576 transgenic mice [132]. Brain-derived Aβ*56 causes memory impairment by injecting to the brains of normal rats [109]. The mechanism by which Aβ*56-induced neurotoxicity is that Aβ*56 mainly binds to NMDAR on neuronal membranes to elevate NMDAR-dependent intracellular Ca^2+^ concentration by increasing Ca^2+^ inflow and activate Ca^2+^-dependent calmodulin kinase IIα (CaMKIIα). The activated CaMKIIα may promote the hyperphosphorylation of tau at Ser202 and Ser416 and trigger AD-related pathological changes [110,111]. Moreover, a recent study suggested that Aβ*56 also damaged ubiquitin-dependent and ubiquitin-independent proteasome functions and promoted cell death [112].

### 4.4. Spherical Oligomers

#### 4.4.1. ASPD

ASPD are spherical oligomers assembled by Aβ40 or Aβ42 with diameter being ∼11.9 ± 1.7 nm in transmission electron microscopic (TEM) images [133,134]. ASPD can be assembled into spherical oligomers with a molecular weight of 330 kDa in units of trimers after 5 h of aggregation in vitro. Even if the reaction time exceeds 24 h, the main fragment in the solution remains to be oligomers without fiber formation, indicating that ASPD aggregation is off-fibril pathway [20]. ASPD do not bind to A11 and OC, indicating that they have a unique conformation different from Aβ dimers, Aβ*56 and other A11-reactive entities [135]. Previous study demonstrated that ASPD had higher neurotoxicity by exerting synaptic damage and inhibiting LTP, but ASPD did not exhibit toxicity to glial cells when they were applied to rat hippocampal slices [113]. After binding to the fourth extracellular loop (Ex4) of Na^+^/K^+^-ATPase α3 subunit (NAKα3) in neurons, brain-derived ASPD cause the impairment of NAKα3-specific activity and the activation of N-type voltage-gated calcium channels, both of which can increase cytoplasmic Ca^2+^ levels and induce mitochondrial calcium dyshomeostasis, leading to mature neurons death and cognitive impairment eventually [20,114]. Further research on the distribution and spread of ASPD among different neurons identified that ASPD mainly accumulated in the trans-Golgi network of excitatory neurons and after secreted by these neurons, ASPD invaded and caused the death of surrounding NAKα3-positive neurons [136]. In general, ASPD are Aβos with unique conformation and robust neurotoxicity.

#### 4.4.2. ADDLs

Aβ-derived diffusible aggregates (ADDLs) are also spherical oligomers that can be detected in AD human brains [137]. The content of ADDLs in brain of Tg2576 mice increases by 5–100 times before a change in spatial memory [138]. ADDLs are small sphere with a diameter of 5–6 nm and can be specifically recognized by the ADDLs-specific antibodies termed as NUx [139]. ADDLs bind to mature neurons, generate neurotoxicity, and induce abnormal autophagy of neurons [115]. In addition, ADDLs can reduce the level of insulin receptor in neurons and disrupt normal insulin signaling [116,117]. They may also bind to EPhB2, trigger the receptor degradation and the phosphorylation of NMDAR, and impair synaptic plasticity [118]. Moreover, ADDLs can lead to abnormal localization of tau to dendrites of primary neurons by activating tyrosine kinase Fyn [119,120]. The latest study showed that after binding to the dendrites and spines of cortical pyramidal neurons, ADDLs induced the rapid depletion of kalirin-7 and caused spine dysgenesis [140].

## 5. Discussion

It is widely accepted that Aβos rather than monomers and fibrils play a critical role in AD progression. Aβos can trigger a series of toxic reactions in neurons, such as receptor disability, mitochondrial damage, Ca^2+^ homeostasis dysregulation and abnormal tau phosphorylation, but the abnormalities of these activities also aggravate accumulation and aggregation of Aβ, forming a vicious cycle. In addition, Aβos are unstable and highly heterogeneous, having various sizes, structures and toxicities both in vivo and in vitro, and play different roles at different phages of AD pathogenesis, resulting in differential consequences on neuronal synaptotoxicity and survival.

Because of the important roles of Aβ in AD pathogenicity, many drugs targeting Aβ have been developed to reduce the production of Aβ or promote the removal of Aβ aggregates and plaque deposits in the brain. Unfortunately, these drugs have failed in the clinical trials due to severe side effects or inability to improve cognitive impairment of AD patients. There may be four reasons for the failure of Aβ-targeted treatment strategies. Firstly, a large number of clinical trials have shown that β-secretase and γ-secretase inhibitors not only fail to effectively improve cognitive impairment of AD patients, but also lead to cognitive deterioration and serious side effects [141,142], which may be attributable to the sharp decrease in the levels of Aβ monomer and secretases, resulting in the inhibition of normal physiological functions of Aβ monomers and secretases. Secondly, many antibodies and vaccines targeting Aβ monomers, oligomers or fibers, such as bapinezumab [143], solanezumab [144] and AN-1792 [145] may not effectively bind to pathogenic oligomer, and usually induce serious side effects. Thirdly, the stage is too late for the tested drugs to interfere with AD development. Aβ aggregates exist in the brain as early as 10–20 years before the occurrence of clinical symptoms, but the current clinical trials of drugs intervene on patients with mild-to-moderate symptomatic. At this stage, a large number of neuronal deaths and irreversible cognitive impairments have occurred in the brains of patients. Finally, AD is a chronic neurodegenerative disease with complex pathological mechanisms, and the exact pathogenesis is currently unclear. In addition to the amyloid cascade hypothesis, researchers also found that tau protein, neuroinflammation and other factors also play important roles in the development of AD. According to the current theoretical basis, drugs just against Aβ may not effectively reverse the pathological processes of AD. In the future, the combination of multiple drugs targeting different pathogenic factors should be considered, which may bring exciting results to AD treatment.

Although significant progress in understanding the biological characteristics of Aβ and the role in the pathogenesis of AD have been made, we have not yet clarified the aggregation kinetics of Aβ, and systematically understood the relationship between different Aβos and AD development due to the heterogeneity and instability of Aβos. Some methods used in the current study to extract and detect Aβos, such as denaturants and SDS-PAGE, may affect the structures and properties of Aβos. In order to obtain high-purity and biologically active Aβos, more gentle methods to separate and purify different Aβos should be explored. In recent years, the acquisition of Aβos has been transformed from incubation with synthesized Aβ to extraction from the brains of AD transgenic mice, making it possible to explore the relationship between pathophysiologic Aβos and AD pathological changes. However, the biological characteristics of Aβos in the brain of AD patients are scarcely studied. Moreover, Aβos exist in the brain long before the occurrence of clinical symptoms, and it is necessary to monitor Aβos in the brain of patients at the preclinical stage to clarify the chronological sequence of different Aβos in AD human, and their detailed toxic mechanisms. More importantly, the structural characteristics of Aβos that play a vital role in AD pathology should be analyzed, and then the therapies against these toxic Aβos could be designed and developed.

## Figures and Tables

**Figure 1 ijms-21-04477-f001:**
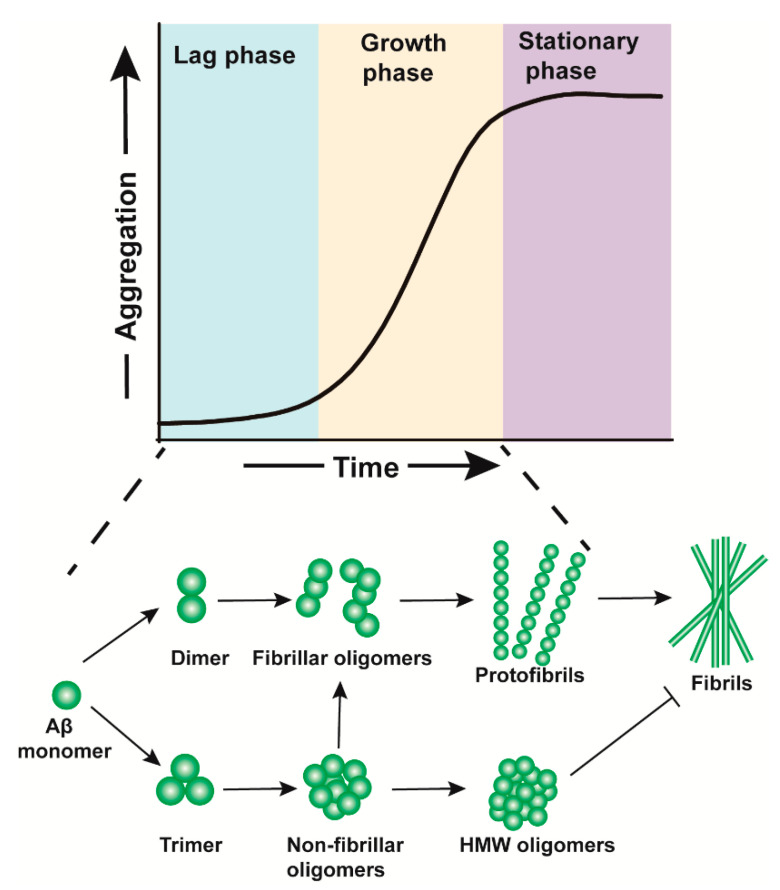
Schematic representation of the process of β-amyloid (Aβ) aggregation. There are three phases from Aβ monomers to mature fibrils: an initial lag phase, a subsequent growth phase and a final stationary phase. β-amyloid oligomers (Aβos), necessary intermediates in fibril generation, mainly exist in lag phase and growth phase. Some Aβos, termed as fibrillar oligomers, can quickly form protofibrils and mature fibrils through fibril-forming pathway, in which the first step corresponds to the formation of Aβ dimers, whereas others such as Aβ trimers or globulomers, termed as non-fibrillar oligomers, are structurally distinct from fibrillar aggregates, and they contribute to the formation of fibrils through a structural conversion to become fibrillar oligomers.

**Figure 2 ijms-21-04477-f002:**
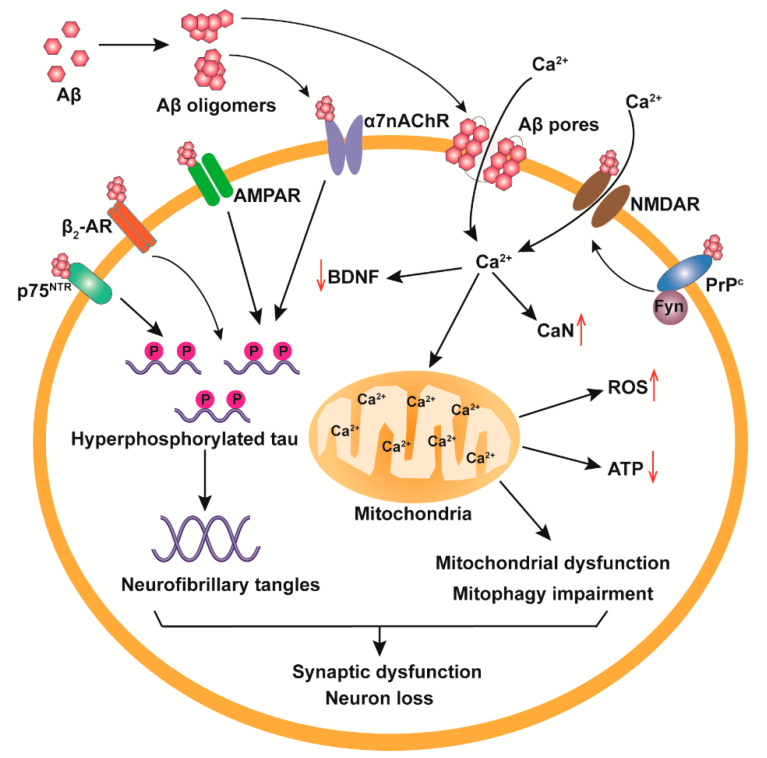
Schematic of the neurotoxicological mechanisms of Aβos. Aβos, mainly produced extracellularly, can bind to multiple receptors on the surface of neuronal membrane and interfere with the normal signaling pathways of the receptors, and they can also directly interact with membrane to form pore structure, leading to the change of membrane integrity and permeability. These two effects of Aβos on cell membrane can further induce Ca^2+^ homeostasis dysregulation, mitochondrial damage, reactive oxygen species (ROS) generation, reduction of ATP level and abnormal tau phosphorylation, resulting in synaptic dysfunction, neuron loss and impairment of long-term potentiation (LTP).

**Table 1 ijms-21-04477-t001:** The characteristics and toxicity of several Aβ oligomers.

Aβos	Structure	Toxicity	The Type of Research	References
Dimers	Covalently coupled by monomers	Reducing neuron length, inhibiting LTP response	Ex vivo	[101]
		Synaptic damage and tau hyperphosphorylation	Ex vivo	[102]
		Reducing hippocampal acetylcholine and serotonin turnover rates, causing neurotransmitter dysfunction and behavioral impairments	In vivo	[103]
		Neuronal overactivation by inhibiting the glutamate reuptake process	In vivo and ex vivo	[104]
Trimers	__	Nerve cell death	In vitro	[105]
		Cognitive impairment	In vivo	[106]
		Reducing KLC1, changing tau conformation and disrupting axonal transport	Ex vivo and in vivo	[107]
		Inhibiting LTP response	Ex vivo	[108]
Aβ*56	__	Memory impairment	In vivo	[109]
		Elevating NMDAR-dependent intracellular Ca^2+^ concentration, activating CaMKIIα, tau hyperphosphorylation	Ex vivo	[110,111]
		Inactivating ubiquitin-dependent and ubiquitin-independent proteasome	In vitro	[112]
ASPD	Spherical oligomers with diameter being ∼11.9 ± 1.7 nm	Synaptic damage and inhibition of LTP	In vivo and ex vivo	[113]
		Impairing NAKα3-specific activity, activating N-type voltage-gated calcium channels, increasing the levels of cytoplasmic Ca^2+^, mitochondrial calcium dyshomeostasis, mature neuron death and cognitive impairment	Ex vivo	[20,114]
ADDLs	Small sphere with a diameter of 5–6 nm	Abnormal autophagy of neurons	Ex vivo	[115]
		Reducing the level of insulin receptor and disrupting normal insulin signaling	In vitro and in vivo	[116,117]
		Inducing the degradation of EPhB2 receptor and the phosphorylation of NMDAR, impairing synaptic plasticity	Ex vivo and in vivo	[118]
		Abnormal localization of tau to dendrites by activating tyrosine kinase Fyn	Ex vivo and in vivo	[119,120]

Aβos: β-amyloid oligomers; LTP: long-term potentiation; KLC1: kinesin-1 light chain; NMDAR: *N*-methyl-d-aspartate receptor; CaMKIIα: Ca^2+^-dependent calmodulin kinase IIα; ASPD: amylospheroids; NAKα3: Na^+^/K^+^-ATPase α3 subunit; ADDLs: Aβ-derived diffusible aggregates.

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
