# Peer review of "The Toxicity and Polymorphism of β-Amyloid Oligomers"

_ijms, 2020, doi:10.3390/ijms21124477_

Round 1

Reviewer 1 Report

Dear Authors, 

I have read  the manuscript "The Toxicity and Polymorphism of β-Amyloid
Oligomers" such a review could help the field to systematize the knowledge about amyloid-beta proteins.

First, I  would suggest defining that is  "Oligomers".

Second, I would suggest providing a table summarizing structure/ toxicity / the type of research in Vivo, ex vivo, in vitro experiments.

Author Response

Response to Reviewer 1 Comments

Point 1: First, I would suggest defining that is "Oligomers".

Response 1: Oligomers are soluble polymers assembled by monomers, some of which are intermediates in fibril generation. Oligomers have different molecular weights and conformations, which results in various oligomeric properties.

Point 2: Second, I would suggest providing a table summarizing structure/ toxicity / the type of research in Vivo, ex vivo, in vitro experiments.

Response 2: We have summarized the structure/ toxicity/ the type of research in Table 1 of the revised manuscript. (Page 6, lines 212-214; Page 7, lines 215-217)

Table 1. The characteristics and toxicity of several Aβ oligomers.

Aβos

Structure

Toxicity

The type of research

References

Dimers

Covalently coupled by monomers

Reducing neuron length, inhibiting LTP response

Ex vivo

108

Synaptic damage and tau hyperphosphorylation

Ex vivo

109

Reducing hippocampal acetylcholine and serotonin turnover rates, causing neurotransmitter dysfunction and behavioral impairments

In vivo

110

Neuronal overactivation by inhibiting the glutamate reuptake process

In vivo and ex vivo

105

Trimers

__

Nerve cell death

In vitro

115

Cognitive impairment

In vivo

116

Reducing KLC1, changing tau conformation and disrupting axonal transport

Ex vivo and in vivo

117

Inhibiting LTP response

Ex vivo

118

Aβ*56

__

Memory impairment

In vivo

114

Elevating NMDAR-dependent intracellular Ca2+ concentration, activating CaMKⅡα, tau hyperphosphorylation

Ex vivo

112, 123

Inactivating ubiquitin-dependent and ubiquitin-independent proteasome

In vitro

124

ASPD

Synaptic damage and inhibition of LTP

In vivo and ex vivo

128

(Continued on next page)

Aβos

Structure

Toxicity

The type of research

References

ASPD

Spherical oligomers with diameter being ∼11.9 ± 1.7 nm

Impairing NAKα3-specific activity, activating N-type voltage-gated calcium channels, increasing the levels of cytoplasmic Ca2+, mitochondrial calcium dyshomeostasis, mature neuron death and cognitive impairment

Ex vivo

20, 129

ADDLs

Small sphere with a diameter of 5–6 nm

Abnormal autophagy of neurons

Ex vivo

134

Reducing the level of insulin receptor and disrupting normal insulin signaling

In vitro and in vivo

135, 136

Inducing the degradation of EPhB2 receptor and the phosphorylation of NMDAR, impairing synaptic plasticity

Ex vivo and in vivo

137

Abnormal localization of tau to dendrites by activating tyrosine kinase Fyn

Ex vivo and in vivo

138, 139

Aβos: β-amyloid oligomers; LTP: long-term potentiation; KLC1: kinesin-1 light chain; NMDAR: N-methyl-D-aspartate receptor; CaMKⅡα: Ca2+-dependent calmodulin kinase Ⅱα; ASPD: amylospheroids; NAKα3: Na+/K+-ATPase α3 subunit; ADDLs: Aβ-derived diffusible aggregates.

Other adjustments:

  1. We adjusted the spacing after the paragraph to 12 pounds. (Page 2, line 64; Page 3, line 83)
  2. We added the abbreviations in parentheses the first time they appear in figure. (Page 2, line 66, line 68; Page 3, lines 89-91; Page 10, line 376)
  3. We replaced "5. Conclusion" with "5. Discussion". (Page 9, line 326)

Reviewer 2 Report

In this work, the latest findings on the characteristics of amyloid-beta are well summarized, and it would be helpful for readers to understand amyloid-beta dynamics and pathological roles in the development and progression of Alzheimer’s disease (AD). I have several suggestions:

  1. Page 5 (lines 163–168) (Compared with … disruption [69].)
    These sentences are confusing. I suggest that the authors describe the findings of the literatures accurately.
  2. It would be true that amyloid-beta are closely implicated in the pathogenesis of AD; however, the past extensive clinical trials targeting amyloid-beta failed/discontinued. How do the authors think about it? I recommend that the authors provide discussion regarding this issue.
  3. Recent findings on the characteristics of amyloid-beta are well written, but the content of directions for future research is small. I recommend that the authors provide more discussion about future research directions on amyloid-beta study.

Author Response

Response to Reviewer 2 Comments

Point 1: Page 5 (lines 163–168) (Compared with … disruption [69].)

These sentences are confusing. I suggest that the authors describe the findings of the literatures accurately.

Response 1: We changed our statement in our revised manuscript. The sentence “Compared with cholesterol-free cell membrane, cholesterol-rich membrane has stronger binding to Aβ, indicating that cholesterol facilitates the interaction between Aβ and the membrane. Cholesterol can increase the membrane rigidity and protect the membrane from Aβos-mediated damage [69, 72]. Consistently, cholesterol removal decreases the association of Aβ with neuronal membranes, and facilitates the membrane perforation and disruption [69].” was changed to “The effect of cholesterol on the toxicity of Aβos to membrane is controversial. Cholesterol can change the structures of Aβos and facilitate the interaction between Aβ and the membrane, on the other hand, cholesterol increases the membrane rigidity and has a protective effect against Aβos-mediated damage [69, 72].” (Page 5, lines 163-166)

Point 2: It would be true that amyloid-beta are closely implicated in the pathogenesis of AD; however, the past extensive clinical trials targeting amyloid-beta failed/discontinued. How do the authors think about it? I recommend that the authors provide discussion regarding this issue.

Response 2: Thanks for your suggestion, we added following discussion to “5. Discussion” section of the revised manuscript. (Page 9, lines 334-353; Page 10, lines 354-355)

Because of the important roles of Aβ in AD pathogenicity, many drugs targeting Aβ have been developed to reduce the production of Aβ or promote the removal of Aβ aggregates and plaque deposits in the brain. Unfortunately, these drugs have failed in the clinical trials due to severe side effects or inability to improve cognitive impairment of AD patients. There may be four reasons for the failure of Aβ-targeted treatment strategies. Firstly, a large number of clinical trials have shown that β-secretase and γ-secretase inhibitors not only fail to effectively improve cognitive impairment of AD patients, but also lead to cognitive deterioration and serious side effects [1, 2], which may be attributable to the sharp decrease in the levels of Aβ monomer and secretases, resulting in the inhibition of normal physiological functions of Aβ monomers and secretases. Secondly, many antibodies and vaccines targeting Aβ monomers, oligomers or fibers, such as bapinezumab [3], solanezumab [4] and AN-1792 [5] may not effectively bind to pathogenic oligomer, and usually induce serious side effects. Thirdly, the stage is too late for the tested drugs to interfere with AD development. Aβ aggregates exist in the brain as early as 10-20 years before the occurrence of clinical symptoms, but the current clinical trials of drugs intervene on patients with mild-to-moderate symptomatic. At this stage, a large number of neuronal deaths and irreversible cognitive impairments have occurred in the brains of patients. Finally, AD is a chronic neurodegenerative disease with complex pathological mechanisms, and the exact pathogenesis is currently unclear. In addition to the amyloid cascade hypothesis, researchers also found that tau protein, neuroinflammation and other factors also play important roles in the development of AD. According to the current theoretical basis, drugs just against Aβ may not effectively reverse the pathological processes of AD. In the future, the combination of multiple drugs targeting different pathogenic factors should be considered, which may bring exciting results to AD treatment.

Point 3: Recent findings on the characteristics of amyloid-beta are well written, but the content of directions for future research is small. I recommend that the authors provide more discussion about future research directions on amyloid-beta study.

Response 3: Thanks for your suggestion, we added following discussion to “5. Discussion” section of the revised manuscript. (Page 10, lines 356-370)

Although significant progress in understanding the biological characteristics of Aβ and the role in the pathogenesis of AD have been made, we have not yet clarified the aggregation kinetics of Aβ, and systematically understood the relationship between different Aβos and AD development due to the heterogeneity and instability of Aβos. Some methods used in the current study to extract and detect Aβos, such as denaturants and SDS-PAGE, may affect the structures and properties of Aβos. In order to obtain high-purity and biologically active Aβos, more gentle methods to separate and purify different Aβos should be explored. In recent years, the acquisition of Aβos has been transformed from incubation with synthesized Aβ to extraction from the brains of AD transgenic mice, making it possible to explore the relationship between pathophysiologic Aβos and AD pathological changes. However, the biological characteristics of Aβos in the brain of AD patients are scarcely studied. Moreover, Aβos exist in the brain long before the occurrence of clinical symptoms, and it is necessary to monitor Aβos in the brain of patients at the preclinical stage to clarify the chronological sequence of different Aβos in AD human, and their detailed toxic mechanisms. More importantly, the structural characteristics of Aβos that play a vital role in AD pathology should be analyzed, and then the therapies against these toxic Aβos could be designed and developed.

Other adjustments:

  1. We adjusted the spacing after the paragraph to 12 pounds. (Page 2, line 64; Page 3, line 83)
  2. We added the abbreviations in parentheses the first time they appear in figure. (Page 2, line 66, line 68; Page 3, lines 89-91; Page 10, line 376)
  3. We replaced "5. Conclusion" with "5. Discussion". (Page 9, line 326)
  4. Because we discussed the future direction of Aβos in detail in "5. Discussion" of the revised manuscript, we deleted the sentence “It is crucial to clarify the temporal sequence of Aβos in AD human brains and detect the effects of different Aβ aggregates on cognitive impairment and their detailed toxicity mechanisms.” (Page 8, lines 330-332)

References

  1. Henley, D., N. Raghavan, R. Sperling, P. Aisen, R. Raman, and G. Romano, Preliminary Results of a Trial of Atabecestat in Preclinical Alzheimer's Disease. New Engl J Med, 2019. 380, 1483-1485.
  2. Doody, R.S., R. Raman, M. Farlow, T. Iwatsubo, B. Vellas, S. Joffe, K. Kieburtz, F. He, X.Y. Sun, R.G. Thomas, et al., A Phase 3 Trial of Semagacestat for Treatment of Alzheimer's Disease. New Engl J Med, 2013. 369, 341-350.
  3. Salloway, S., R. Sperling, N.C. Fox, K. Blennow, W. Klunk, M. Raskind, M. Sabbagh, L.S. Honig, A.P. Porsteinsson, S. Ferris, et al., Two Phase 3 Trials of Bapineuzumab in Mild-to-Moderate Alzheimer's Disease. New Engl J Med, 2014. 370, 322-333.
  4. Honig, L.S., B. Vellas, M. Woodward, M. Boada, R. Bullock, M. Borrie, K. Hager, N. Andreasen, E. Scarpini, H. Liu-Seifert, et al., Trial of Solanezumab for Mild Dementia Due to Alzheimer's Disease. New Engl J Med, 2018. 378, 321-330.
  5. Nicoll, J.A.R., G.R. Buckland, C.H. Harrison, A. Page, S. Harris, S. Love, J.W. Neal, C. Holmes, and D. Boche, Persistent neuropathological effects 14 years following amyloid-beta immunization in Alzheimer's disease. Brain, 2019. 142, 2113-2126.

Round 2

Reviewer 1 Report

Dear Authors,  I have no more comments.

Author Response

  Thank you for your very helpful comments and suggestions. We have incorporated the suggestions one point by one in this revised manuscript and believe that your suggestions have significantly improved our manuscript, and hope that the revision we have made will address your concerns.